# RetroBridge: Modeling Retrosynthesis with Markov Bridges

**Ilia Igashov**[*]
École Polytechnique Fédérale de Lausanne
`ilia.igashov@epfl.ch`

**Arne Schneuing**[*]
École Polytechnique Fédérale de Lausanne
`arne.schneuing@epfl.ch`

**Marwin Segler**
Microsoft Research
`marwinsegler@microsoft.com`

**Michael Bronstein**
University of Oxford
`michael.bronstein@cs.ox.ac.uk`

**Bruno Correia**
École Polytechnique Fédérale de Lausanne
`bruno.correia@epfl.ch`

## Abstract

Retrosynthesis planning is a fundamental challenge in chemistry which aims at designing multi-step reaction pathways from commercially available starting materials to a target molecule. Each step in multi-step retrosynthesis planning requires accurate prediction of possible precursor molecules given the target molecule and confidence estimates to guide heuristic search algorithms. We model single-step retrosynthesis as a distribution learning problem in a discrete state space. First, we introduce the Markov Bridge Model, a generative framework aimed to approximate the dependency between two intractable discrete distributions accessible via a finite sample of coupled data points. Our framework is based on the concept of a Markov bridge, a Markov process pinned at its endpoints. Unlike diffusion-based methods, our Markov Bridge Model does not need a tractable noise distribution as a sampling proxy and directly operates on the input product molecules as samples from the intractable prior distribution. We then address the retrosynthesis planning problem with our novel framework and introduce RetroBridge, a template-free retrosynthesis modeling approach that achieves state-of-the-art results on standard evaluation benchmarks.

## 1 Introduction

Computational and machine learning methods for *de novo* drug design show great promise as more cost-effective alternatives to experimental high-throughput screening approaches [Thomas et al., 2023] to propose molecules with desirable properties. While *in silico* results suggest high predicted target binding affinities and other favorable properties of the generated molecules, limited emphasis has so far been placed on their synthesizability [Stanley and Segler, 2023]. For laboratory testing, synthetic pathways need to be developed for the newly designed molecules, which is an extremely challenging and time-consuming task.

Retrosynthesis planning [Corey, 1991, Strieth-Kalthoff et al., 2020, Tu et al., 2023] tools address this challenge by proposing reaction steps or entire pathways that can be validated and optimized in the

---

[*]These authors contributed equally

NeurIPS 2023 AI for Science Workshop.

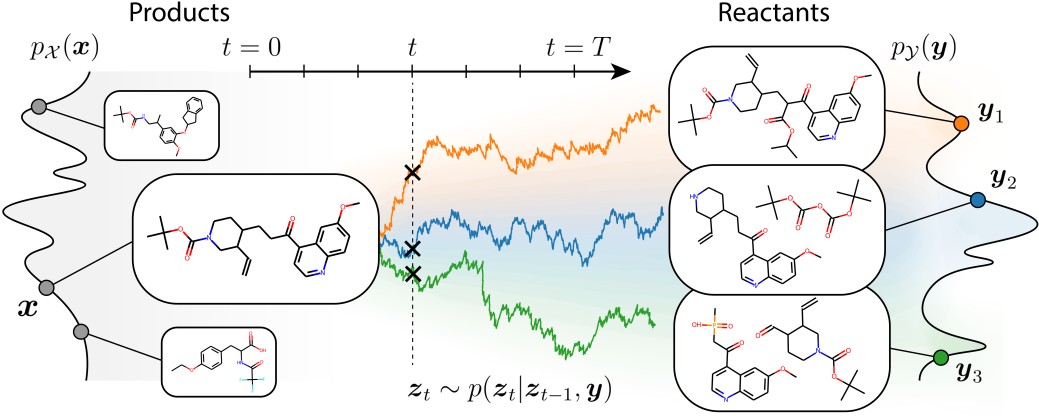

Figure 1: Markov bridges between the distribution of products and distribution of reactants.

lab. Single-step retrosynthesis models predict precursor molecules for a given target molecule [Segler and Waller, 2017, Coley et al., 2017, Liu et al., 2017, Strieth-Kalthoff et al., 2020, Tu et al., 2023]. Applying these methods recursively allows to decompose the initial molecule in progressively simpler intermediates and eventually reach available starting molecules [Segler et al., 2018].

While most works have used a discriminative formulation for retrosynthesis modeling [Strieth-Kalthoff et al., 2020, Tu et al., 2023, Jiang et al., 2022], we propose to view the task as a conditional distribution learning problem, as shown in Figure 1. This approach has several advantages, including the ability to model uncertainty and to generate new and diverse retrosynthetic pathways. Furthermore, and most importantly, the probabilistic formulation reflects the fact that the same product molecule can often be synthesized with different sets of reactants and reagents.

Diffusion models [Sohl-Dickstein et al., 2015, Ho et al., 2020] and other modern score-based and flow-based generative methods [Rezende and Mohamed, 2015, Song et al., 2020, Lipman et al., 2022, Albergo and Vanden-Eijnden, 2022, Albergo et al., 2023] may seem like good candidates for retrosynthesis modeling. However, as we show in this work, such models do not fit naturally to the formulation of the problem, as they are designed to approximate a single intractable data distribution. To do so, one typically samples initial noise from a simple prior distribution and then maps it to a data point that follows a complex target distribution. In contrast, we aim to learn the dependency between two intractable distributions rather than one intractable distribution itself. While this can be achieved by conditioning the sampling process on the relevant context and keep sampling from the prior noise, we show here that such use of the original unconditional generative idea is suboptimal for approximating the dependency between two discrete distributions.

In this work, we propose RetroBridge, a template-free probabilistic method for single-step retrosynthesis modeling. As shown in Figure 1, we model the dependency between the spaces of products and reactants as a stochastic process that is constrained to start and to end at specific data points. To this end, we introduce the Markov Bridge Model, a generative model that learns the dependency between two intractable discrete distributions through the finite sample of coupled data points. Taking a product molecule as input, our method models the trajectories of Markov bridges starting at the given product and ending at data points following the distribution of reactants. To score reactant graphs sampled in this way, we leverage the probabilistic nature of RetroBridge and measure its uncertainty at each sample. We demonstrate that RetroBridge achieves competitive results on standard retrosynthesis modeling benchmarks. Besides, we compare RetroBridge with the state-of-the-art graph diffusion model DiGress [Vignac et al., 2022], and demonstrate quantitatively and qualitatively that the proposed Markov Bridge Model is better suited to tasks where two intractable discrete distributions need to be mapped.

To summarise, the main contributions of this work are the following:

- We introduce the Markov Bridge Model to approximate the probabilistic dependency between two intractable discrete distributions accessible via a finite sample of coupled data points.
- We demonstrate the superiority of the proposed formulation over diffusion models in the context of learning the dependency between two intractable discrete distributions.
- We propose RetroBridge, the first Markov Bridge Model for retrosynthesis modeling. RetroBridge is a template-free single-step retrosynthesis prediction method that achieves state-of-the-art results on standard benchmarks.

## 2 Related Work

**Diffusion Models** Diffusion models [Sohl-Dickstein et al., 2015, Ho et al., 2020] form a class of powerful and effective score-based generative methods that have recently achieved promising results in many different domains including protein design [Watson et al., 2023], small molecule generation [Hoogeboom et al., 2022, Igashov et al., 2022, Schneuing et al., 2022], molecular docking [Corso et al., 2022], and sampling of transition state molecular structures [Duan et al., 2023, Kim et al., 2023]. While most models are designed for the continuous data domain, a few methods were proposed to operate on discrete data [Hoogeboom et al., 2021, Johnson et al., 2021, Austin et al., 2021, Yang et al., 2023] and, in particular, on discrete graphs [Vignac et al., 2022]. To the best of our knowledge, however, no diffusion models have been applied to modeling chemical reactions and recovering retrosynthetic pathways.

**Schrödinger Bridge Problem** Given two distributions and a reference stochastic process between them, solving the Schrödinger bridge (SB) problem [Schrödinger, 1932, Léonard, 2013] amounts to finding a process closest to the reference in terms of Kullback-Leibler divergence on path spaces. While most recent methods employ the SB formalism in the context of unconditional generative modeling [Vargas et al., 2021, Wang et al., 2021, De Bortoli et al., 2021, Chen et al., 2021, Bunne et al., 2023, Liu et al., 2022], a few works aimed to approximate the reference stochastic process through training on coupled samples from two continuous distributions [Holdijk et al., 2022, Somnath et al., 2023]. To the best of our knowledge, there are no methods operating on categorical distributions, which is the subject of the present work.

**Retrosynthesis Modeling** Recent retrosynthesis prediction methods can be divided into two main groups: template-based and template-free methods [Jiang et al., 2022]. While template-based methods depend on predefined sets of specific reaction templates or leaving groups, template-free methods are less restricted and therefore are able to explore new reaction pathways. Two common data representations used for retrosynthesis prediction are symbolic representations SMILES [Weininger, 1988] and molecular graphs. A variety of language models have been recently proposed [Liu et al., 2017, Zheng et al., 2019, Tetko et al., 2020] to operate on SMILES. Due to the nature of the sequence-to-sequence translation problem, all these methods are template-free. Among the existing graph-based methods [Segler and Waller, 2017], the most recent template-based ones are GLN [Dai et al., 2019], GraphRetro [Somnath et al., 2021] and LocalRetro [Chen and Jung, 2021], and template-free approaches are G2G [Shi et al., 2020] and MEGAN [Sacha et al., 2021]. In this work, we propose a novel template-free graph-based method.

## 3 RetroBridge

We frame the retrosynthesis prediction task as a generative problem of modeling a stochastic process between two discrete-valued distributions of products $p_{\mathcal{X}}$ and reactants $p_{\mathcal{Y}}$. These distributions are intractable and are represented by a finite collection of $D$ coupled samples $\{(\boldsymbol{x}_i, \boldsymbol{y}_i)\}_{i=1}^{D}$, where $\boldsymbol{x}_i \sim p_{\mathcal{X}}(\boldsymbol{x}_i)$ is a product molecule and $\boldsymbol{y}_i \sim p_{\mathcal{Y}}(\boldsymbol{y}_i)$ is a corresponding set of reactant molecules. While products and reactants follow distributions $p_{\mathcal{X}}$ and $p_{\mathcal{Y}}$ respectively, there is a dependency between these variables that can be expressed in the form of the joint distribution $p_{\mathcal{X},\mathcal{Y}}$ such that $\int p_{\mathcal{X},\mathcal{Y}}(\boldsymbol{x}, \boldsymbol{y}) d\boldsymbol{x} = p_{\mathcal{Y}}(\boldsymbol{y})$ and $\int p_{\mathcal{X},\mathcal{Y}}(\boldsymbol{x}, \boldsymbol{y}) d\boldsymbol{y} = p_{\mathcal{X}}(\boldsymbol{x})$. The joint distribution $p_{\mathcal{X},\mathcal{Y}}$ is also intractable and accessible only through the discrete sample of coupled data points $\{(\boldsymbol{x}_i, \boldsymbol{y}_i)\}_{i=1}^{D}$.

First, we introduce the Markov Bridge Model, a general framework for learning the dependency between two intractable discrete-valued distributions. Next, we discuss a special case where random

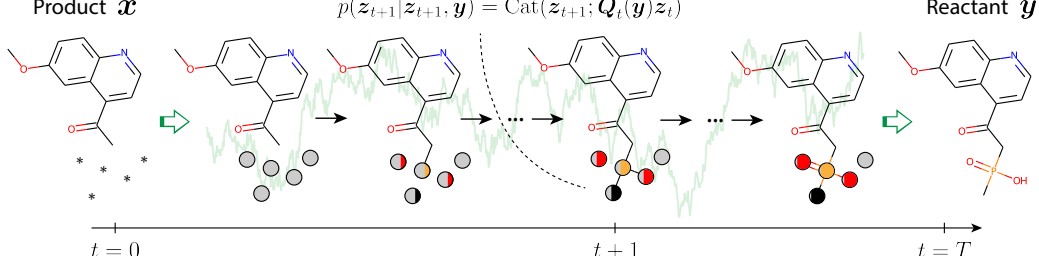

**Figure 2:** The process of changing atom types along the trajectory of the Markov bridge. The trajectory starts at time step $t = 0$ with the product molecule and several disconnected "dummy" atoms that will be included in the final reactant molecule. The probability of sampling the target atom type increases as $t$ grows. Five circles filled with different colors represent these probabilities. To make the illustration less bulky, we omitted a part of the product molecule and one of two reactant molecules.

variables are molecular graphs. Upon this formulation, we introduce RetroBride, a Markov Bridge Model for single-step retrosynthesis modeling. Finally, we explain a simple but rather effective way of scoring RetroBridge samples based on the statistical uncertainty of the model.

## 3.1 Markov Bridge Model

We model the dependency between two discrete spaces $\mathcal{X}$ and $\mathcal{Y}$ by a Markov bridge [Fitzsimmons et al., 1992, Çetin and Danilova, 2016], which is a Markov process pinned to specific data points in the beginning and in the end. For a pair of samples $(\boldsymbol{x}, \boldsymbol{y}) \sim p_{\mathcal{X}, \mathcal{Y}}(\boldsymbol{x}, \boldsymbol{y})$ and a sequence of time steps $t = 0, 1, \ldots, T$, we define the corresponding Markov bridge as a sequence of random variables $(\boldsymbol{z}_t)_{t=0}^T$, that starts at $\boldsymbol{x}$, i.e., $\boldsymbol{z}_0 = \boldsymbol{x}$, and satisfies the Markov property,

$$p(\boldsymbol{z}_t | \boldsymbol{z}_0, \boldsymbol{z}_1, \ldots, \boldsymbol{z}_{t-1}, \boldsymbol{y}) = p(\boldsymbol{z}_t | \boldsymbol{z}_{t-1}, \boldsymbol{y}). \tag{1}$$

To pin the process at the data point $\boldsymbol{y}$, we introduce an additional requirement,

$$p(\boldsymbol{z}_T = \boldsymbol{y} | \boldsymbol{z}_{T-1}, \boldsymbol{y}) = 1. \tag{2}$$

Assuming that both distributions $p_{\mathcal{X}}$ and $p_{\mathcal{Y}}$ are categorical with a finite sample space $\{1, \ldots, K\}$, we can represent data points as $K$-dimensional one-hot vectors: $\boldsymbol{x}, \boldsymbol{y}, \boldsymbol{z}_t \in \mathbb{R}^K$. To model a Markov bridge defined by equations (1-2), similar to Austin et al. [2021], we introduce a sequence of transition matrices $\boldsymbol{Q}_0, \boldsymbol{Q}_1, \ldots, \boldsymbol{Q}_{T-1} \in \mathbb{R}^{K \times K}$, defined as

$$\boldsymbol{Q}_t := \boldsymbol{Q}_t(\boldsymbol{y}) = \alpha_t \boldsymbol{I}_K + (1 - \alpha_t) \boldsymbol{y} \boldsymbol{1}_K^\top, \tag{3}$$

where $\boldsymbol{I}_K$ is a $K \times K$ identity matrix, $\boldsymbol{1}_K$ is a $K$-dimensional all-one vector, and $\alpha_t$ is a schedule parameter transitioning from $\alpha_0 = 1$ to $\alpha_{T-1} = 0$. Transition probabilities (1) can be written as follows,

$$p(\boldsymbol{z}_{t+1} | \boldsymbol{z}_t, \boldsymbol{y}) = \mathrm{Cat}\left(\boldsymbol{z}_{t+1}; \boldsymbol{Q}_t \boldsymbol{z}_t\right), \tag{4}$$

where $\mathrm{Cat}(\cdot\,; \boldsymbol{p})$ is a categorical distribution with probabilities given by $\boldsymbol{p}$. We note that setting $\alpha_{T-1} = 0$ ensures the requirement (2).

Using the finite set of coupled samples $\{(\boldsymbol{x}_i, \boldsymbol{y}_i)\}_{i=1}^D \sim p_{\mathcal{X}, \mathcal{Y}}$, our goal is to learn a Markov bridge (1-2) to be able to sample $\boldsymbol{y}$ when only $\boldsymbol{x}$ is available. To do this, we replace $\boldsymbol{y}$ with an approximation $\hat{\boldsymbol{y}}$ computed with a neural network $\varphi_\theta$:

$$\hat{\boldsymbol{y}} = \varphi_\theta(\boldsymbol{z}_t, t), \tag{5}$$

and define an approximated transition kernel,

$$q_\theta(\boldsymbol{z}_{t+1} | \boldsymbol{z}_t) = \mathrm{Cat}\left(\boldsymbol{z}_{t+1}; \boldsymbol{Q}_t(\hat{\boldsymbol{y}}) \boldsymbol{z}_t\right). \tag{6}$$

---

**Algorithm 1** Training of the Markov Bridge Model

---

**Input:** coupled sample $(\boldsymbol{x}, \boldsymbol{y}) \sim p_{\mathcal{X}, \mathcal{Y}}$, neural network $\varphi_\theta$

$t \sim \mathcal{U}(0, \ldots, T-1),\ \boldsymbol{z}_t \sim \text{Cat}\left(\boldsymbol{z}_t; \overline{\boldsymbol{Q}}_{t-1}\boldsymbol{x}\right)$   ▷ Sample time step and intermediate state

$\hat{\boldsymbol{y}} \leftarrow \varphi_\theta(\boldsymbol{z}_t, t)$   ▷ Output of $\varphi_\theta$ is a vector of probabilities

$p(\boldsymbol{z}_{t+1}|\boldsymbol{z}_t, \boldsymbol{y}) \leftarrow \text{Cat}\left(\boldsymbol{z}_{t+1}; \boldsymbol{Q}_t(\boldsymbol{y})\boldsymbol{z}_t\right)$   ▷ Reference transition distribution

$q_\theta(\boldsymbol{z}_{t+1}|\boldsymbol{z}_t) \leftarrow \text{Cat}\left(\boldsymbol{z}_{t+1}; \boldsymbol{Q}_t(\hat{\boldsymbol{y}})\boldsymbol{z}_t\right)$   ▷ Approximated transition distribution

Minimize $D_{\text{KL}}\left(p(\boldsymbol{z}_{t+1}|\boldsymbol{z}_t, \boldsymbol{y})\|q_\theta(\boldsymbol{z}_{t+1}|\boldsymbol{z}_t)\right)$

---

---

**Algorithm 2** Sampling

---

**Input:** starting point $\boldsymbol{x} \sim p_{\mathcal{X}}$, neural network $\varphi_\theta$

$\boldsymbol{z}_0 \leftarrow \boldsymbol{x}$

**for** $t$ in $0, ..., T-1$:

  $\hat{\boldsymbol{y}} \leftarrow \varphi_\theta(\boldsymbol{z}_t, t)$   ▷ Output of $\varphi_\theta$ is a vector of probabilities

  $q_\theta(\boldsymbol{z}_{t+1}|\boldsymbol{z}_t) \leftarrow \text{Cat}\left(\boldsymbol{z}_{t+1}; \boldsymbol{Q}_t(\hat{\boldsymbol{y}})\boldsymbol{z}_t\right)$   ▷ Approximated transition distribution

  $\boldsymbol{z}_{t+1} \sim q_\theta(\boldsymbol{z}_{t+1}|\boldsymbol{z}_t)$

Return $\boldsymbol{z}_T$

---

We train $\varphi_\theta$ by maximizing a lower bound of log-likelihood $\log q_\theta(\boldsymbol{y}|\boldsymbol{x})$. As shown in Appendix A.1, it has the following closed-form expression,

$$\log q_\theta(\boldsymbol{y}|\boldsymbol{x}) \geq -T \cdot \mathbb{E}_{t \sim \mathcal{U}(0,\ldots,T-1)} \mathbb{E}_{\boldsymbol{z}_t \sim p(\boldsymbol{z}_t|\boldsymbol{x}, \boldsymbol{y})} D_{\text{KL}}\left(p(\boldsymbol{z}_{t+1}|\boldsymbol{z}_t, \boldsymbol{y})\|q_\theta(\boldsymbol{z}_{t+1}|\boldsymbol{z}_t)\right). \quad (7)$$

For any $\boldsymbol{x} \in \mathcal{X}, \boldsymbol{y} \in \mathcal{Y}$, and $t = 1, \ldots, T$, sampling of $\boldsymbol{z}_t$ can be effectively performed using a cumulative product matrix $\overline{\boldsymbol{Q}}_t = \boldsymbol{Q}_t \boldsymbol{Q}_{t-1}...\boldsymbol{Q}_0$. As shown in Appendix A.2, the cumulative matrix $\overline{\boldsymbol{Q}}_t$ can be written in closed form,

$$\overline{\boldsymbol{Q}}_t = \overline{\alpha}_t \boldsymbol{I}_K + (1 - \overline{\alpha}_t)\boldsymbol{y}\boldsymbol{1}_K^\top, \quad (8)$$

where $\overline{\alpha}_t = \prod_{s=0}^t \alpha_s$. Therefore, $p(\boldsymbol{z}_{t+1}|\boldsymbol{z}_0, \boldsymbol{z}_T)$ can be written as follows,

$$p(\boldsymbol{z}_{t+1}|\boldsymbol{z}_0, \boldsymbol{z}_T) = \text{Cat}\left(\boldsymbol{z}_{t+1}; \overline{\boldsymbol{Q}}_t \boldsymbol{z}_0\right). \quad (9)$$

To sample a data point $\boldsymbol{y} \equiv \boldsymbol{z}_T$ starting from a given $\boldsymbol{z}_0 \equiv \boldsymbol{x} \sim p_{\mathcal{X}}(\boldsymbol{x})$, one iteratively predicts $\hat{\boldsymbol{y}} = \varphi_\theta(\boldsymbol{z}_t, t)$ and then derives $\boldsymbol{z}_{t+1} \sim q_\theta(\boldsymbol{z}_{t+1}|\boldsymbol{z}_t) = \text{Cat}\left(\boldsymbol{z}_{t+1}; \boldsymbol{Q}_t(\hat{\boldsymbol{y}})\boldsymbol{z}_t\right)$ for $t = 0, \ldots, T-1$. Training and sampling procedures of the Markov Bridge Model are provided in Algorithms 1 and 2 respectively.

### 3.2 RetroBridge: Markov Bridge Model for Retrosynthesis Planning

In our setup, each data point is a molecular graph with nodes representing atoms and edges corresponding to covalent bonds. We represent the molecular graph with a matrix of node features $\boldsymbol{H} \in \mathbb{R}^{N \times K_a}$ which are, for instance, one-hot encoded atom types, and a tensor of edge features $\boldsymbol{E} \in \mathbb{R}^{N \times N \times K_e}$, which can be one-hot encoded bond types.

In the scope of our probabilistic framework, we consider such a molecular graph representation as a collection of independent categorical random variables. More formally, we denote product and reactants data points $\boldsymbol{x}$ and $\boldsymbol{y}$ as tuples of the corresponding node and edge feature tensors: $\boldsymbol{x} = [\boldsymbol{H}_x, \boldsymbol{E}_x]$ and $\boldsymbol{y} = [\boldsymbol{H}_y, \boldsymbol{E}_y]$. For such complex data points, we modify the definitions of transition matrices and probabilities accordingly:

$$[\boldsymbol{Q}_t^H]_j = \alpha_t \boldsymbol{I}_{K_a} + (1 - \alpha_t)\boldsymbol{1}_{K_a}[\boldsymbol{H}_y]_j, \qquad p(\boldsymbol{H}_{t+1}|\boldsymbol{H}_t, \boldsymbol{H}_y) = \text{Cat}\left(\boldsymbol{H}_{t+1}; \boldsymbol{H}_t \boldsymbol{Q}_t^H\right), \quad (10)$$

$$[\boldsymbol{Q}_t^E]_{j,k} = \alpha_t \boldsymbol{I}_{K_e} + (1 - \alpha_t)\boldsymbol{1}_{K_e}[\boldsymbol{E}_y]_{j,k}, \qquad p(\boldsymbol{E}_{t+1}|\boldsymbol{E}_t, \boldsymbol{E}_y) = \text{Cat}\left(\boldsymbol{E}_{t+1}; \boldsymbol{E}_t \boldsymbol{Q}_t^E\right), \quad (11)$$

where $[\boldsymbol{H}]_j \in \mathbb{R}^{1 \times K_a}$ is the $j$-th row of the feature matrix $\boldsymbol{H}$ (i.e. transposed feature vector of the $j$-th node), and $[\boldsymbol{E}]_{j,k} \in \mathbb{R}^{1 \times K_e}$ is the transposed feature vector of the edge between $j$-th and $k$-th nodes.

Because some atoms present in the reactant molecules can be absent in the corresponding product molecule, we add "dummy" nodes to the initial graph of the product. As shown in Figure 2, some "dummy" nodes are transformed into atoms of reactant molecules. In our experiments, we always add 10 "dummy" nodes to the initial product graphs.

## 3.3 Confidence and Scoring

It is important to have a reliable scoring method that selects the most relevant sets of reactants out of all generated samples. In order to rank RetroBridge samples, we benefit from the probabilistic nature of the model and utilize its confidence in the generated samples as a scoring function. We estimate the confidence of the model by computing the likelihood $q_\theta(\boldsymbol{y}|\boldsymbol{x})$ of a set of reactants $\boldsymbol{y}$ for an input product molecule $\boldsymbol{x}$. For a set of $M$ samples $\{\boldsymbol{z}_T^{(i)}\}_{i=1}^M$ generated by RetroBridge for an input product $\boldsymbol{x}$, we compute the likelihood-based confidence score for the set of reactants $\boldsymbol{y}$ as follows,

$$q_\theta(\boldsymbol{y}|\boldsymbol{x}) = \mathbb{E}_{\boldsymbol{y}' \sim q_\theta(\cdot|\boldsymbol{x})} \mathbb{1}\{\boldsymbol{y}' = \boldsymbol{y}\} \approx \frac{1}{M} \sum_{i=1}^M \mathbb{1}\{\boldsymbol{z}_T^{(i)} = \boldsymbol{y}\}. \tag{12}$$

# 4 Results

## 4.1 Experimental Setup

**Dataset**   For all the experiments we use the USPTO-50k dataset [Schneider et al., 2016] which includes 50k reactions found in the US patent literature. We use standard train/validation/test splits provided by Dai et al. [2019]. Somnath et al. [2021] report that the dataset contains a shortcut in that the product atom with atom-mapping 1 is part of the edit in almost 75% of the cases. Even though our model does not depend on the order of graph nodes, we utilize the dataset version with canonical SMILES provided by Somnath et al. [2021]. Besides, we randomly permute graph nodes once SMILES are read and converted to graphs.

**Baselines**   We compare RetroBridge with template-based methods GLN [Dai et al., 2019], Local-Retro [Chen and Jung, 2021], and GraphRetro [Somnath et al., 2021], and template-free methods MEGAN [Sacha et al., 2021], G2G [Shi et al., 2020], Augmented Transformer Tetko et al. [2020] and SCROP [Zheng et al., 2019]. We note that SCROP, G2G and Augmented Transformer were trained and evaluated using different data splits provided by Liu et al. [2017]. We obtained GLN predictions using the publicly available code and model weights[2] and used the latest LocalRetro predictions provided by its authors. Additionally, MEGAN was originally trained and evaluated on random data splits, so we retrained and evaluated it ourselves. Finally, as described in Appendix A.4, we compare RetroBridge with the state-of-the-art discrete graph diffusion model DiGress Vignac et al. [2022] and a template-free baseline based on a graph transformer architecture [Dwivedi and Bresson, 2020, Vignac et al., 2022].

**Evaluation**   For each input product, we sample 100 reactant sets and report top-$k$ exact match accuracy ($k = 1, 3, 5, 10$) which is measured as the proportion of input products for which the method managed to produce the correct set of reactants in its top-$k$ samples. Subsequently, for top-$k$ samples produced for every input product, we run the forward reaction prediction model Molecular Transformer [Schwaller et al., 2019] and report round-trip accuracy and coverage [Schwaller et al., 2020]. Round-trip accuracy is the percentage of correctly predicted reactants among all predictions, where reactants are considered correct either if they match the ground truth or if they lead back to the input product. Round-trip coverage, on the other hand, measures if there is at least one correct prediction in the top-$k$ according to the definition above. These metrics reflect the fact that one product can be mapped to multiple different valid sets of reactants, as shown in Figure 1.

## 4.2 Neural Network

We use a graph transformer network [Dwivedi and Bresson, 2020, Vignac et al., 2022] to approximate the final state of the Markov bridge process. We represent molecules as fully-connected graphs where node features are one-hot encoded atom types (sixteen atom types and additional "dummy" type) and edge features are covalent bond types (three bond types and additional "none" type). Similarly to Vignac et al. [2022] we compute several graph-related node and global features that include number of cycles and spectral graph features. Details on the network architecture, hyperparameters and training process are provided in Appendix A.3.

---

[2]Note that the top-$k$ exact match accuracies differ from the one originally reported values because we deduplicate outputs for our evaluation.

Table 1: Top-$k$ accuracy (exact match) on the USPTO-50k test dataset. Methods marked with asterisk were trained and evaluated on different data splits reported by Liu et al. [2017]. The best performing template-based (TB) and template-free (TF) methods are highlighted in bold.

| | Model | $k = 1$ | $k = 3$ | $k = 5$ | $k = 10$ |
|---|---|---|---|---|---|
| TB | GLN [Dai et al., 2019] | 52.5 | 74.7 | 81.2 | 87.9 |
| | GraphRetro [Somnath et al., 2021] | **53.7** | 68.3 | 72.2 | 75.5 |
| | LocalRetro [Chen and Jung, 2021] | 52.6 | **76.0** | **84.4** | **90.6** |
| TF | SCROP* [Zheng et al., 2019] | 43.7 | 60.0 | 65.2 | 68.7 |
| | G2G* [Shi et al., 2020] | 48.9 | 67.6 | 72.5 | 75.5 |
| | Aug. Transformer* [Tetko et al., 2020] | 48.3 | — | 73.4 | 77.4 |
| | MEGAN [Sacha et al., 2021] | 48.0 | 70.9 | 78.1 | 85.4 |
| | RetroBridge (ours) | **50.8** | **74.1** | **80.6** | **85.6** |

Table 2: Top-$k$ round-trip coverage and accuracy on the USPTO-50k test dataset. The best performing template-based (TB) and template-free (TF) methods are highlighted in bold.

| | | Coverage | | | Accuracy | | |
|---|---|---|---|---|---|---|---|
| | Model | $k = 1$ | $k = 3$ | $k = 5$ | $k = 1$ | $k = 3$ | $k = 5$ |
| TB | GLN [Dai et al., 2019] | **82.5** | 92.0 | 94.0 | **82.5** | 71.0 | 66.2 |
| | LocalRetro [Chen and Jung, 2021] | 82.1 | **92.3** | **94.7** | 82.1 | 71.0 | **66.7** |
| TF | MEGAN [Sacha et al., 2021] | 78.1 | 88.6 | 91.3 | 78.1 | 67.3 | 61.7 |
| | RetroBridge (ours) | **85.1** | **95.7** | **97.1** | **85.1** | **73.6** | **67.8** |

## 4.3 Retrosynthesis Modeling

Here we report top-$k$ and round-trip accuracy for RetroBridge and other state-of-the-art methods on the USPTO-50k test set. Table 1 provides exact match accuracy results, and Table 2 reports round-trip accuracy computed using Molecular Transformer [Schwaller et al., 2019].

For completeness, we compared exact match accuracy results of RetroBrdige with both template-free and template-based methods. However, template-free modeling is a more challenging task, and RetroBridge is template-free, therefore we primarily focus on the latter group of methods. As shown in Table 1, RetroBridge outperforms other template-free methods, in particular highly optimized transformer-based models.

Because exact match accuracy cannot reflect the complete picture of dependencies between spaces of products and reactants, we additionally measure round-trip results using an orthogonal method for forward reaction prediction. We evaluate predictions of the template-based methods GLN [Dai et al., 2019] and LocalRetro [Chen and Jung, 2021] as well as the retrained version of template-free MEGAN [Sacha et al., 2021] ourselves for comparison. The results are reported in Table 2. RetroBridge clearly outperforms the template-free baseline and, in spite of a much higher difficulty of the template-free setup, even achieves higher round-trip coverage and accuracy values than state-of-the-art template-based methods. These results support our hypothesis that retrosynthesis should be modeled in a probabilistic framework considering the absence of a unique set of reactants for a given initial product molecule.

## 4.4 Examples

Figure 3 shows three examples of reactions we randomly selected from the USPTO-50k test set. For each of these examples, we provide top-3 RetroBridge samples and the corresponding confidence scores. In all three cases our model managed to recover the correct set of reactants. In the first case, RetroBridge predicts the correct reactants with a high confidence. In this case, the gap between the scores of the first and the second prediction is remarkably high (0.66 vs 0.12). In both other cases, the model is not as confident in the answer. This uncertainty is reflected in the scores: top-1 samples

Figure 3: Examples of modeled reactants. We selected three random inputs from the test set and for each of them we provide the top-3 RetroBridge predictions along with their confidence scores. Two check marks indicate that sampled reactants are the same as the ground truth, and one check mark means that reactants are different, but Molecular Transformer [Schwaller et al., 2019] predicts the product molecule used as input.

(which are not correct) have scores 0.18 and 0.38 respectively (cf. 0.17 and 0.2 for the correct ones). More examples are provided in Figure 5.

## 5   Conclusion

In this work, we introduce the Markov Bridge Model, a new generative framework for tasks that involve learning dependencies between two intractable discrete-valued distributions. We furthermore apply the new methodology to the retrosynthesis prediction problem, which is an important challenge in medicinal chemistry and drug discovery. Our template-free method, RetroBridge, achieves state-of-the-art results on common evaluation benchmarks. Importantly, our experiments show that choosing a suitable probabilistic modeling framework positively affects the performance on this task compared to the straightforward adaptation of diffusion models.

While this work is focused on the retrosynthesis modeling task, we note that application of Markov Bridge Models is not limited to this problem. The proposed framework can be used in many other settings where two discrete distributions accessible via a finite sample of coupled data points need to be mapped. Such applications include but are not limited to image-to-image translation, inpainting, text translation and design of protein binders. We leave exploration of Markov Bridge Models in the scope of these and other possible challenges for future work.

### Acknowledgments

We thank Max Welling, Philippe Schwaller, Rebecca Neeser, Clément Vignac and Anar Rzayev for helpful feedback and insightful discussions. Ilia Igashov has received funding from the European Union's Horizon 2020 research and innovation programme under the Marie Skłodowska-Curie grant agreement No 945363. Arne Schneuing is supported by Microsoft Research AI4Science.

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

# A  Appendix

## A.1  Variational Lower Bound

The log-likelihood of reactants $\boldsymbol{y} \equiv \boldsymbol{z}_T$ given product $\boldsymbol{x} \equiv \boldsymbol{z}_0$ can be written as follows,

$$\log q_\theta(\boldsymbol{y}|\boldsymbol{x}) = \log q_\theta(\boldsymbol{z}_T|\boldsymbol{z}_0) \tag{13}$$

$$= \log \int d\boldsymbol{z}_{1:T-1} q_\theta(\boldsymbol{z}_{1:T}|\boldsymbol{z}_0) \tag{14}$$

$$= \log \int d\boldsymbol{z}_{1:T-1} \prod_{t=1}^{T} q_\theta(\boldsymbol{z}_t|\boldsymbol{z}_{t-1}) \tag{15}$$

$$= \log \int d\boldsymbol{z}_{1:T-1} \frac{p(\boldsymbol{z}_{1:T}|\boldsymbol{z}_0, \boldsymbol{z}_T)}{p(\boldsymbol{z}_{1:T}|\boldsymbol{z}_0, \boldsymbol{z}_T)} \prod_{t=1}^{T} q_\theta(\boldsymbol{z}_t|\boldsymbol{z}_{t-1}) \tag{16}$$

$$= \log \int d\boldsymbol{z}_{1:T-1} p(\boldsymbol{z}_{1:T}|\boldsymbol{z}_0, \boldsymbol{z}_T) \prod_{t=1}^{T} \frac{q_\theta(\boldsymbol{z}_t|\boldsymbol{z}_{t-1})}{p(\boldsymbol{z}_t|\boldsymbol{z}_{t-1}, \boldsymbol{z}_T)}. \tag{17}$$

Using Jensen's inequality (JI) and the fact that $p(\boldsymbol{z}_{1:T}|\boldsymbol{z}_0, \boldsymbol{z}_T) = p(\boldsymbol{z}_{1:T-1}|\boldsymbol{z}_0, \boldsymbol{z}_T)$ $(*)$ we can derive a lower bound of this log-likelihood,

$$\log q_\theta(\boldsymbol{y}|\boldsymbol{x}) \overset{\text{JI}}{\geq} \int d\boldsymbol{z}_{1:T-1} p(\boldsymbol{z}_{1:T}|\boldsymbol{z}_0, \boldsymbol{z}_T) \log \prod_{t=1}^{T} \frac{q_\theta(\boldsymbol{z}_t|\boldsymbol{z}_{t-1})}{p(\boldsymbol{z}_t|\boldsymbol{z}_{t-1}, \boldsymbol{z}_T)} \tag{18}$$

$$\overset{(*)}{=} \int d\boldsymbol{z}_{1:T-1} p(\boldsymbol{z}_{1:T-1}|\boldsymbol{z}_0, \boldsymbol{z}_T) \log \prod_{t=1}^{T} \frac{q_\theta(\boldsymbol{z}_t|\boldsymbol{z}_{t-1})}{p(\boldsymbol{z}_t|\boldsymbol{z}_{t-1}, \boldsymbol{z}_T)} \tag{19}$$

$$= \sum_{t=1}^{T} \int d\boldsymbol{z}_{1:T-1} p(\boldsymbol{z}_{1:T-1}|\boldsymbol{z}_0, \boldsymbol{z}_T) \log \frac{q_\theta(\boldsymbol{z}_t|\boldsymbol{z}_{t-1})}{p(\boldsymbol{z}_t|\boldsymbol{z}_{t-1}, \boldsymbol{z}_T)} \tag{20}$$

$$= \mathcal{L}_1(\boldsymbol{z}_0, \boldsymbol{z}_T) + \sum_{t=2}^{T} \mathcal{L}_t(\boldsymbol{z}_0, \boldsymbol{z}_T). \tag{21}$$

Here, the first term $\mathcal{L}_1$ can be written as follows,

$$\mathcal{L}_1(\boldsymbol{z}_0, \boldsymbol{z}_T) = \int d\boldsymbol{z}_1 p(\boldsymbol{z}_1|\boldsymbol{z}_0, \boldsymbol{z}_T) \log \frac{q_\theta(\boldsymbol{z}_1|\boldsymbol{z}_0)}{p(\boldsymbol{z}_1|\boldsymbol{z}_0, \boldsymbol{z}_T)} = -D_{\text{KL}}\left(p(\boldsymbol{z}_1|\boldsymbol{z}_0, \boldsymbol{z}_T)\|q_\theta(\boldsymbol{z}_1|\boldsymbol{z}_0)\right). \tag{22}$$

Using Bayes' rule (BR) and the Markov propery (MP) of the Markov bridge $p$, we can derive a similar expression for all intermediate terms $\mathcal{L}_t$,

$$\mathcal{L}_t(\boldsymbol{z}_0, \boldsymbol{z}_T) = \int d\boldsymbol{z}_{t-1} d\boldsymbol{z}_t p(\boldsymbol{z}_{t-1}, \boldsymbol{z}_t|\boldsymbol{z}_0, \boldsymbol{z}_T) \log \frac{q_\theta(\boldsymbol{z}_t|\boldsymbol{z}_{t-1})}{p(\boldsymbol{z}_t|\boldsymbol{z}_{t-1}, \boldsymbol{z}_T)} \tag{23}$$

$$\overset{\text{BR}}{=} \int d\boldsymbol{z}_{t-1} d\boldsymbol{z}_t p(\boldsymbol{z}_{t-1}|\boldsymbol{z}_0, \boldsymbol{z}_T) p(\boldsymbol{z}_t|\boldsymbol{z}_{t-1}, \boldsymbol{z}_0, \boldsymbol{z}_T) \log \frac{q_\theta(\boldsymbol{z}_t|\boldsymbol{z}_{t-1})}{p(\boldsymbol{z}_t|\boldsymbol{z}_{t-1}, \boldsymbol{z}_T)} \tag{24}$$

$$\overset{\text{MP}}{=} \int d\boldsymbol{z}_{t-1} d\boldsymbol{z}_t p(\boldsymbol{z}_{t-1}|\boldsymbol{z}_0, \boldsymbol{z}_T) p(\boldsymbol{z}_t|\boldsymbol{z}_{t-1}, \boldsymbol{z}_T) \log \frac{q_\theta(\boldsymbol{z}_t|\boldsymbol{z}_{t-1})}{p(\boldsymbol{z}_t|\boldsymbol{z}_{t-1}, \boldsymbol{z}_T)} \tag{25}$$

$$= -\int d\boldsymbol{z}_{t-1} p(\boldsymbol{z}_{t-1}|\boldsymbol{z}_0, \boldsymbol{z}_T) D_{\text{KL}}\left(p(\boldsymbol{z}_t|\boldsymbol{z}_{t-1}, \boldsymbol{z}_T)\|q_\theta(\boldsymbol{z}_t|\boldsymbol{z}_{t-1})\right). \tag{26}$$

We combine $\mathcal{L}_1$ and $\mathcal{L}_t$ to obtain the final expression for the variational lower bound of the log-likelihood:

$$\log q_\theta(\boldsymbol{y}|\boldsymbol{x}) \geq -\sum_{t=1}^{T} \mathbb{E}_{\boldsymbol{z}_{t-1} \sim p(\boldsymbol{z}_{t-1}|\boldsymbol{x}, \boldsymbol{y})} D_{\text{KL}}\left(p(\boldsymbol{z}_t|\boldsymbol{z}_{t-1}, \boldsymbol{y})\|q_\theta(\boldsymbol{z}_t|\boldsymbol{z}_{t-1})\right) \tag{27}$$

$$= -\sum_{t=0}^{T-1} \mathbb{E}_{\boldsymbol{z}_t \sim p(\boldsymbol{z}_t|\boldsymbol{x}, \boldsymbol{y})} D_{\text{KL}}\left(p(\boldsymbol{z}_{t+1}|\boldsymbol{z}_t, \boldsymbol{y})\|q_\theta(\boldsymbol{z}_{t+1}|\boldsymbol{z}_t)\right). \tag{28}$$

Finally, we obtain the form provided in (7) by replacing the sum over all $T$ terms with its unbiased estimator,

$$\log q_\theta(\boldsymbol{y}|\boldsymbol{x}) \geq -T \cdot \mathbb{E}_{t\sim\mathcal{U}(0,...,T-1)}\mathbb{E}_{\boldsymbol{z}_t\sim p(\boldsymbol{z}_t|\boldsymbol{x},\boldsymbol{y})} D_{\text{KL}}\left(p(\boldsymbol{z}_{t+1}|\boldsymbol{z}_t,\boldsymbol{y})\|q_\theta(\boldsymbol{z}_{t+1}|\boldsymbol{z}_t)\right). \tag{29}$$

## A.2 Cumulative Transition Matrix $\bar{Q}_t$

A proof by induction. For $t = 0$, by definition, $\overline{\boldsymbol{Q}}_0 = \boldsymbol{Q}_0$ and $\overline{\alpha}_0 = \alpha_0$.

Assume that for $t > 0$ Equation 8 holds, i.e. $\overline{\boldsymbol{Q}}_t = \overline{\alpha}_t \boldsymbol{I}_K + (1 - \overline{\alpha}_t)\boldsymbol{y}\boldsymbol{1}_K^\top$ and $\overline{\alpha}_t = \prod_{s=0}^t \alpha_s$. Then for $t + 1$ we have

$$\overline{\boldsymbol{Q}}_{t+1} = \boldsymbol{Q}_{t+1}\overline{\boldsymbol{Q}}_t \tag{30}$$
$$= \left[\alpha_{t+1}\boldsymbol{I}_K + (1 - \alpha_{t+1})\boldsymbol{y}\boldsymbol{1}_K^\top\right]\left[\overline{\alpha}_t\boldsymbol{I}_K + (1 - \overline{\alpha}_t)\boldsymbol{y}\boldsymbol{1}_K^\top\right] \tag{31}$$
$$= \alpha_{t+1}\overline{\alpha}_t\boldsymbol{I}_K + (\alpha_{t+1} - \alpha_{t+1}\overline{\alpha}_t + \overline{\alpha}_t - \alpha_{t+1}\overline{\alpha}_t)\boldsymbol{y}\boldsymbol{1}_K^\top \tag{32}$$
$$+ (1 - \alpha_{t+1} - \overline{\alpha}_t + \alpha_{t+1}\overline{\alpha}_t)\boldsymbol{y}\boldsymbol{1}_K^\top\boldsymbol{y}\boldsymbol{1}_K^\top. \tag{33}$$

Note that $\boldsymbol{y}\boldsymbol{1}_K^\top\boldsymbol{y}\boldsymbol{1}_K^\top = \boldsymbol{y}\boldsymbol{1}_K^\top$ and $\alpha_{t+1}\overline{\alpha}_t = \overline{\alpha}_{t+1}$. Therefore, we get

$$\overline{\boldsymbol{Q}}_{t+1} = \overline{\alpha}_{t+1}\boldsymbol{I}_K + (\alpha_{t+1} - \overline{\alpha}_{t+1} + \overline{\alpha}_t - \overline{\alpha}_{t+1} + 1 - \alpha_{t+1} - \overline{\alpha}_t + \overline{\alpha}_{t+1})\boldsymbol{y}\boldsymbol{1}_K^\top \tag{34}$$
$$= \overline{\alpha}_{t+1}\boldsymbol{I}_K + (1 - \overline{\alpha}_{t+1})\boldsymbol{y}\boldsymbol{1}_K^\top. \tag{35}$$

## A.3 Implementation Details

### A.3.1 Noise schedule

In all experiments, we use the cosine schedule [Nichol and Dhariwal, 2021]

$$\alpha_t = \cos\left(0.5\pi\frac{t/T + s}{1 + s}\right)^2 \tag{36}$$

with $s = 0.008$ and number of time steps $T = 500$.

### A.3.2 Additional Features

We represent molecules as fully-connected graphs where node features are one-hot encoded atom types (sixteen atom types and additional "dummy" type) and edge features are covalent bond types (three bond types and additional "none" type). Besides, we use a global graph feature $\boldsymbol{y}$ which includes the normalized time step: $\boldsymbol{y} = t/T$. Similarly to Vignac et al. [2022] we compute additional node features. For completeness, we provide the description of these features as in [Vignac et al., 2022] below.

**Cycles**   Rings of different sizes are crucial features of many bioactive molecules but graph neural networks are unable to detect them [Chen et al., 2020]. We therefore add both global cycle counts $\boldsymbol{y}_k$, that capture the overall number of cycles in the graph, as well as local cycle counts $\boldsymbol{H}_k$, that measure how many cycles each node belongs to. These quantities are computed for $k$-cycles up to size $k = 5$ and $k = 6$ for local and global counts, respectively. As proposed by Vignac et al. [2022][3], we use the

---

[3]We use a corrected equation for $\boldsymbol{H}_5$.

following equations that can be efficiently computed on GPUs

$$\boldsymbol{H}_3 = \frac{1}{2}\operatorname{diag}(\boldsymbol{A}^3)$$

$$\boldsymbol{H}_4 = \frac{1}{2}\left(\operatorname{diag}(\boldsymbol{A}^4) - \boldsymbol{d}\odot(\boldsymbol{d}-\mathbf{1}_n) - \boldsymbol{A}\boldsymbol{d}\right)$$

$$\boldsymbol{H}_5 = \frac{1}{2}\left(\operatorname{diag}(\boldsymbol{A}^5) - 2\boldsymbol{T}\boldsymbol{d} - 2\boldsymbol{d}\operatorname{diag}(\boldsymbol{A}^3) - \boldsymbol{A}\operatorname{diag}(\boldsymbol{A}^3) + 5\operatorname{diag}(\boldsymbol{A}^3)\right)$$

$$\boldsymbol{y}_3 = \frac{1}{3}\boldsymbol{H}_3^\top\mathbf{1}_n$$

$$\boldsymbol{y}_4 = \frac{1}{4}\boldsymbol{H}_4^\top\mathbf{1}_n$$

$$\boldsymbol{y}_5 = \frac{1}{5}\boldsymbol{H}_5^\top\mathbf{1}_n$$

$$\boldsymbol{y}_6 = \operatorname{Tr}(\boldsymbol{A}^6) - 3\operatorname{Tr}(\boldsymbol{A}^3\odot\boldsymbol{A}^3) + 9\|\boldsymbol{A}(\boldsymbol{A}^2\odot\boldsymbol{A}^2)\|_F - 6\operatorname{diag}(\boldsymbol{A}^2)^\top\operatorname{diag}(\boldsymbol{A}^4)$$
$$+ 6\operatorname{Tr}(\boldsymbol{A}^4) - 4\operatorname{Tr}(\boldsymbol{A}^3) + 4\operatorname{Tr}(\boldsymbol{A}^2\boldsymbol{A}^2\odot\boldsymbol{A}^2) + 3\|\boldsymbol{A}^3\|_F - 12\operatorname{Tr}(\boldsymbol{A}^2\odot\boldsymbol{A}^2) + 4\operatorname{Tr}(\boldsymbol{A}^2)$$

where $\boldsymbol{A}\in\mathbb{R}^{n\times n}$ is the adjacency matrix and $\boldsymbol{d}\in\mathbb{R}^n$ the node degree vector. Matrix $\boldsymbol{T} = \boldsymbol{A}\odot\boldsymbol{A}^2$ indicates how many triangles each nodes shares with every other node.

**Spectral features**  Again following [Vignac et al., 2022], we include spectral features based on the eigenvalues and eigenvectors of the graph Laplacian. We use the multiplicity of eigenvalue 0 and the first five nonzero eigenvalues as graph-level features, and an indicator of the largest connected component (approximated based on the eigenvectors corresponding to zero eigenvalues) as well as two eigenvectors corresponding to the first nonzero eigenvalues as node-level features.

**Molecular features**  We also tried adding the molecular weight as a graph-level feature and each atom's valency as node-level features, following [Vignac et al., 2022]. Models trained with these features were used everywhere except experiments reported in Tables 1 and 2. Later on, we found out that removing these features improves the performance on the validation set. Therefore, the final model from Tables 1 and 2 does not use molecular features.

### A.3.3  Neural Network Architecture

We use a graph transformer network [Dwivedi and Bresson, 2020, Vignac et al., 2022] to approximate the final state of the Markov bridge process. The architecture of the network is provided in Figure 4A. First, node, edge and global features are passed through an encoder which is implemented as an MLP. Next, the encoded features are processed by a sequence of Graph Transformer Layers (GTL). As shown in Figure 4B, GTL first updates node features using self-attention and combines its output with edge features via FiLM [Perez et al., 2018]:

$$\operatorname{FiLM}(\boldsymbol{X}_1, \boldsymbol{X}_2) = \boldsymbol{X}_1\boldsymbol{W}_1 + (\boldsymbol{X}_1\boldsymbol{W}_2)\odot\boldsymbol{X}_2 + \boldsymbol{W}_2, \tag{37}$$

where $\boldsymbol{W}_1$ and $\boldsymbol{W}_2$ are learnable parameters. Then, edge features are updated using attention scores and global features. To update the global features, GTL combines encoded global features and node and edge features aggregated with PNA:

$$\operatorname{PNA}(\boldsymbol{X}) = \operatorname{concat}(\max(\boldsymbol{X}), \min(\boldsymbol{X}), \operatorname{mean}(\boldsymbol{X}), \operatorname{std}(\boldsymbol{X}))\boldsymbol{W}, \tag{38}$$

where $\boldsymbol{W}$ is a learnable parameter.

Finally, to obtain the final graph representation, updated node and edge features are passed through the decoder which is implemented as an MLP.

### A.3.4  Training

We train our models on a single GPU Tesla V100-PCIE-32GB using AdamW optimizer [Loshchilov and Hutter, 2017] with learning rate $0.0002$ and batch size $64$. We trained models for up to 1000 epochs (which takes several days) and then selected the best checkpoints based on top-5 accuracy (that was computed on a subset of the USPTO-50k validation set).

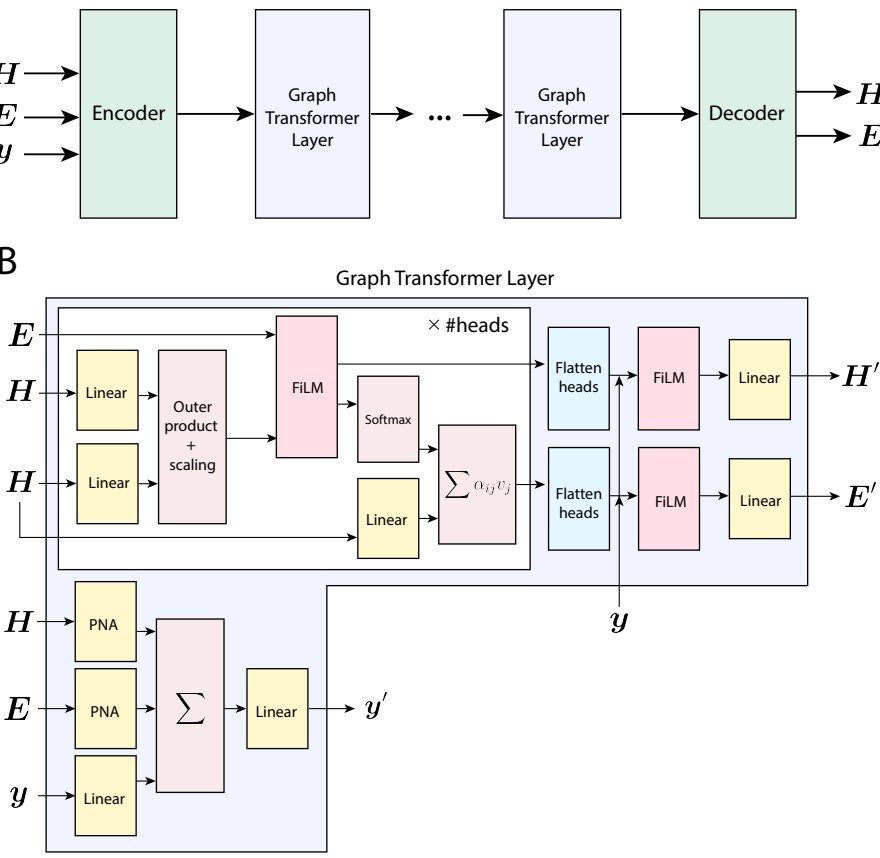

Figure 4: Architecture of the network that approximates the final state of the Markov bridge process (A) and scheme of the Graph Transformer Layer (B).

### A.4 Additional Experiments

In this section, we compare RetroBridge with the naïve adaptation of DiGress [Vignac et al., 2022] for retrosynthesis prediction, which can be considered the most comparable diffusion-based method for this task, and with a graph transformer network that predicts reactants in a one-shot fashion. Furthermore, we study the effect of adding an input product molecule as context to the neural network $\varphi_\theta$ at each sampling step. More precisely, for models with no context we use the formulation (5), while models with context compute predictions as $\hat{y} = \varphi_\theta(z_t, x, t)$. Finally, we try a simpler cross-entropy (CE) loss function, as used in DiGress, that directly compares approximated reactants with the ground-truth,

$$\mathcal{L}_{\mathrm{CE}}(\theta) = T \cdot \mathbb{E}_{t \sim \mathcal{U}(0,\dots,T-1)} \mathbb{E}_{z_t, z_T \sim p} \mathrm{CrossEntropy}(z_T, \varphi_\theta(z_t, t)). \tag{39}$$

In all experiments we use the same neural network architectures and sets of hyperparameters. We perform our evaluation on the USPTO-50k validation set and report top-$k$ accuracy of the generated samples in Table 3.

First of all, we observe that iterative sampling as performed by diffusion models or the Markov Bridge Model is essential for solving the problem of mapping between two graph distributions. Our one-shot graph transformer model trained for a comparable amount of time does not manage to recover any of the reactants. Indeed, it is an extremely challenging task as even a single incorrectly predicted bond or atom type is detrimental for the accuracy metric. To the best of our knowledge, all one-shot graph-based models proposed for retrosynthesis prediction fall into the category of template-based methods. In this case, the networks do not predict the entire set of reactants right away, but instead

Table 3: Additional experiments: top-$k$ accuracy on the USPTO-50k validation set.

| Model | $k = 1$ | $k = 3$ | $k = 5$ | $k = 10$ | $k = 50$ |
|---|---|---|---|---|---|
| DiGress (context) | 47.32 | 68.56 | 73.93 | 78.45 | 80.88 |
| RetroBridge-CE (no context) | 48.71 | 66.84 | 72.33 | 76.08 | 79.38 |
| RetroBridge-CE (context) | **50.74** | 71.50 | 76.58 | 79.50 | 80.58 |
| RetroBridge-VLB (no context) | 47.42 | 69.46 | 75.21 | 79.40 | 83.82 |
| RetroBridge-VLB (context) | 48.92 | **73.04** | **79.44** | **83.74** | **86.31** |

aim to solve much simpler tasks such as prediction of graph edits. To obtain the final set of reactants, graph edit predictions should be further processed by an additional block that typically relies on the predefined reaction templates or leaving group dictionaries.

Next, we compare RetroBridge with DiGress to demonstrate that the Markov bridge formulation is more suitable than diffusion models when two discrete graph distributions are to be mapped. As shown in Table 3, RetroBridge outperforms DiGress with context in all metrics. We note that if we do not pass the context, DiGress predictably does not manage to recover any reactants. This result illustrates that the Markov bridge framework captures the underlying structure of the task much more naturally than diffusion models. A diffusion model maps sampled noise to the reactants having access to the input product molecule only through the additional context while the Markov bridge model starts each sampling trajectory with a product molecule from the intractable distribution $p_{\mathcal{X}}(\boldsymbol{x}) = \int p_{\mathcal{X},\mathcal{Y}}(\boldsymbol{x}, \boldsymbol{y}) d\boldsymbol{y}$.

Finally, we demonstrate that the variational lower bound loss (7) works better than a simplified cross-entropy loss (39) proposed by Vignac et al. [2022]. Ultimately, we find it beneficial to include the input product as context at each sampling step. Unlike the diffusion approach, however, RetroBridge achieves reasonable accuracy values even without additional context as large parts of the product structure are retained throughout the sampling trajectory.

### A.5 Forward Reaction Prediction

We additionally trained and evaluated two models for forward reaction prediction using USPTO-50k and USPTO-MIT datasets. We used the same hyperparameters as in other experiments. As shown in Table 4, our models (ForwardBridge) demonstrate comparable performance with other state-of-the-art methods. However, we stress that the probabilistic formulation is less applicable to the forward reaction prediction task, and under certain assumptions this problem can be considered as completely deterministic. Therefore, we leave the study of capabilities of the Markov Bridge Model in the context of forward reaction prediction out of the scope of this work.

Table 4: Top-$k$ accuracy for forward reaction prediction.

| Dataset | Model | $k = 1$ | $k = 3$ | $k = 5$ |
|---|---|---|---|---|
| USPTO-50k | ForwardBridge (ours) | 89.9 | 93.9 | 94.0 |
| USPTO-MIT | ForwardBridge (ours) | 81.6 | 88.5 | 89.8 |
| | MEGAN | 86.3 | 92.4 | 94.0 |
| | Mol. Transformer | 88.7 | 93.1 | 94.2 |

Figure 5: Examples of modeled reactants. We selected 10 random inputs from the USPTO-50k test set and for each of them we provide the top-3 RetroBridge predictions along with their confidence scores. Two check marks indicate that sampled reactants are the same as the ground truth, and one check mark means that reactants are different, but Molecular Transformer [Schwaller et al., 2019] predicts the product molecule used as input.

