# OpenReview forum: "RetroBridge: Modeling Retrosynthesis with Markov Bridges"
_NeurIPS.cc/2023/Workshop/AI4Science — NeurIPS2023-AI4Science Poster_

### Official Review · Reviewer_XbGS · 2023-10-21
**State of the art retrosynthesis method**

**Rating:** 7
**Confidence:** 4

**Review:**

This paper introduces a novel method for one-step retrosynthesis using probabilistic template-free models. The approach outperforms other template-free methods on single-step retrosynthesis and ostensibly performs better for round-trip accuracy. Overall, this is a useful contribution that I would like to see further investigated in a full paper. I have a few questions that it would be nice have clarified in the paper.
- I slightly confused about the K-dimensional one-hot vectors mentioned. What do these one-hot vectors actually represent?
- Why do you think Retrobridge is better at round-trip accuracy than the other template-free and template-based methods? It would be nice to have some more analysis of this in the paper.
- What is the time-complexity of inferences for this method, and how does that compare to the other methods evaluated?

---

### Official Review · Reviewer_t2Xf · 2023-10-25
**This work introduces the Markov Bridge Model to approximate dependencies between two discrete distributions using sampled data points, showcases its superiority over diffusion models, and presents RetroBridge, a state-of-the-art, template-free method for single-step retrosynthesis prediction.**

**Rating:** 8
**Confidence:** 4

**Review:**

## Pros

* The derivation of the formulas is clear, and the figures are easy to understand.
* They have introduced the Markov Bridge into Retrosynthesis Planning to characterize the distribution of reactant and product, which is novel.
* Ample experiments were conducted and showed comparisons across multiple benchmarks, significantly outperforming other template-free models in single-step retrosynthesis prediction.
* Additional experiments illustrate that the Markov Bridge framework captures the underlying structure of the task much more naturally than diffusion models(DiGress).

## Cons

* Possibly due to method limitations, this paper only discussed template-free prediction for `Reaction type unknown`, but did not discuss the case of `Reaction type given as prior`, like in MEGAN's Table 2 [https://arxiv.org/pdf/2006.15426.pdf](https://arxiv.org/pdf/2006.15426.pdf).
* The discussion in Appendix A.5 and Table 4 regarding the poor performance of the forward reaction prediction model (Forward Bridge) adapted using this method is confusing. Since chemical reactions and Markov processes can both be considered reversible processes, the performance of Forward Bridge and Retro Bridge should be positively correlated. Authors may want to discuss more about the unexpected performance differences.

## Quality

Despite the issues raised in the cons, this is a good paper. The derivations and experiments are sufficient to substantiate its conclusions.

## Clarity

Good

## Originality

No issues

---

### Meta-Review · Area_Chair_x9LR · 2023-10-26

**Recommendation:** Accept (Poster)
**Confidence:** 5

**Metareview:**

This paper models retrosynthesis prediction as a distribution learning problem. Given the distribution of product, it aims to learn the conditional distribution of reactants. Although this paper employs a clear notation for the formulation, the proposed method is not novel given MEGAN.